



# A decadal satellite record of gravity wave activity in the lower stratosphere to study polar stratospheric cloud formation

Lars Hoffmann[1], Reinhold Spang[2], Andrew Orr[3], M. Joan Alexander[4], Laura A. Holt[4], and Olaf Stein[1]

[1]Jülich Supercomputing Centre, Forschungszentrum Jülich, Jülich, Germany
[2]Institut für Energie- und Klimaforschung, Forschungszentrum Jülich, Jülich, Germany
[3]British Antarctic Survey, NERC, Cambridge, UK
[4]NorthWest Research Associates, Inc., CoRA Office, Boulder, CO, USA

*Correspondence to:* L. Hoffmann (l.hoffmann@fz-juelich.de)

**Abstract.** Atmospheric gravity waves yield substantial small-scale temperature fluctuations that can trigger the formation of polar stratospheric clouds (PSCs). This paper introduces a new satellite record of gravity wave activity in the polar lower stratosphere to investigate this process. The record comprises observations of the Atmospheric InfraRed Sounder (AIRS) aboard NASA's Aqua satellite during January 2003 to December 2012. Gravity wave activity is measured in terms of detrended
and noise-corrected $15\,\mu$m brightness temperature variances, which are calculated from AIRS channels that are most sensitive to temperature fluctuations at about $17-32\,$km altitude. The analysis of temporal patterns in the data set revealed a strong seasonal cycle in wave activity with wintertime maxima at mid and high latitudes. The analysis of spatial patterns indicated that orography as well as jet and storm sources are the main cause of the observed waves. Wave activity is closely correlated with $30\,$hPa zonal winds, which is attributed to the AIRS observational filter. We used the new data set to evaluate explicitly
resolved temperature fluctuations due to gravity waves in the European Centre for Medium-Range Weather Forecast (ECMWF) operational analysis. It was found that the analysis reproduces orographic and non-orographic wave patterns in the right places, but that wave amplitudes are typically underestimated by a factor of $2-3$. Furthermore, in a first survey of joint AIRS and Michelson Interferometer for Passive Atmospheric Sounding (MIPAS) satellite observations nearly 50 gravity wave-induced PSC formation events were identified. The survey shows that the new AIRS data set can help to better identify such events and
more generally highlights the importance of the process for polar ozone chemistry.

## 1 Introduction

Polar stratospheric clouds (PSCs) play a key role in ozone chemistry in the polar lower stratosphere. The particles provide the surface area for heterogeneous reactions that catalyze the conversion of chlorine from reservoir gases like $ClONO_2$ and $HCl$ into active chlorine radicals, which accelerate ozone loss in the polar lower stratosphere in late winter and spring (Solomon
et al., 1986; Solomon, 1999). Furthermore, the sedimentation of large $HNO_3$-containing cloud particles leads to denitrification of the stratosphere, which in turn decreases $NO_2$ concentrations, increases $ClO/ClONO_2$ ratios, and accelerates chlorine-catalyzed ozone loss (Toon et al., 1986). PSC particles are typically classified into three different types (Poole and McCormick, 1988; Browell et al., 1990). This includes metastable hydrates of $HNO_3$ such as nitric acid trihydrate (NAT) (Hanson and



Mauersberger, 1988; Worsnop et al., 1993), supercooled ternary solution (STS) droplets of $H_2SO_4/HNO_3/H_2O$ (Zhang et al., 1993; Carslaw et al., 1994), and water ice (Steele et al., 1983). Thermodynamic analyses indicate that NAT and STS can exist at temperatures well above the frost point of ice, with $T_{ice} < T_{STS} < T_{NAT}$. Typical PSC formation and existence temperatures at 20 km altitude are $T_{ice} = 188\,\text{K}$, $T_{STS} = 191\,\text{K}$, and $T_{NAT} = 195\,\text{K}$ (Pawson et al., 1995). However, actual values vary with

atmospheric composition and altitude.

     Atmospheric gravity waves are an important driver of middle atmosphere dynamics and have a substantial impact on weather and climate. They transport energy and momentum from lower to upper altitudes, contribute to turbulence and mixing, and influence the mean circulation and thermal structure of the middle atmosphere (Lindzen, 1981; Holton, 1982). Gravity waves are triggered by a variety of sources. The most prominent sources are orography (Smith, 1985; Durran and Klemp, 1987;

Nastrom and Fritts, 1992) and convection (Pfister et al., 1986; Tsuda et al., 1994; Alexander and Pfister, 1995). Other sources include adjustment of unbalanced flows near jet streams and frontal systems (Fritts and Alexander, 2003; Wu and Zhang, 2004). Here we are interested in gravity waves because they can provide significant local temperature fluctuations that can trigger the formation of PSCs, even if synoptic-scale temperatures are above formation thresholds. Case studies of mountain waves induced by the Scandinavian mountains showed that the waves can cause localized cooling of up to $10-15\,\text{K}$ (Carslaw

et al., 1998b; Dörnbrack et al., 1999, 2002). The Antarctic Peninsula is another well-known hotspot for the formation of PSCs from mountain waves in the southern hemisphere (Höpfner et al., 2006b; Eckermann et al., 2009; Orr et al., 2015). In the Antarctic polar stratosphere wave-induced PSC formation is particularly important in fall or spring, whereas synoptic-scale temperatures in winter are usually well below the PSC formation threshold (Campbell and Sassen, 2008; McDonald et al., 2009; Noel and Pitts, 2012). The Arctic stratospheric vortex is generally warmer and more disturbed due to planetary wave

activity, making gravity waves possibly an even more important source of PSCs in the northern hemisphere (Alexander et al., 2013). Although most case studies focus on gravity waves from orographic sources, Hitchman et al. (2003) and Shibata et al. (2003) showed that non-orographic gravity waves can also trigger PSC formation.

     Previous studies on PSC formation used mesoscale model output (Höpfner et al., 2006a; Noel and Pitts, 2012; Orr et al., 2015) or global positioning system radio occultation observations (Kohma and Sato, 2011; Alexander et al., 2011, 2013) as

a source of information on gravity waves. The use of Atmospheric InfraRed Sounder (AIRS) observations for that purpose was first explored by Eckermann et al. (2009) and Lambert et al. (2012). In this paper we present a new AIRS data set of gravity wave activity in the polar lower stratosphere that can be used to study the impact of gravity wave-induced temperature fluctuations on PSC formation in more detail. The data set is based on a ten-year record (January 2003 to December 2012) of observations and includes more than $5.3 \times 10^9$ infrared radiance spectra measured by AIRS at mid and high latitudes during

that time. As a measure of gravity wave activity the new data set provides detrended and noise-corrected $15\,\mu\text{m}$ brightness temperature variances on a $4° \times 2°$ horizontal grid on a daily basis. The AIRS channels selected for the data product provide most sensitivity in the lower and mid stratosphere (about $17-32\,\text{km}$), i. e. in the altitude range that is most relevant for PSC formation. Here we used the new AIRS data set to identify local hotspots and sources of gravity wave activity, to characterize its seasonal cycle at northern and southern mid and high latitudes, and to analyze correlations with stratospheric background

winds.





We exploited the new AIRS data set in two applications. Various studies demonstrated that AIRS measurements are particularly suited to validate high-resolution gravity wave simulations. Some studies focused on the validation of convective waves (Kim et al., 2009; Grimsdell et al., 2010; Stephan and Alexander, 2015; Wu et al., 2015), but case studies for mountain waves have also been shown (Orr et al., 2015). The validation of explicitly resolved temperature fluctuations in high-resolution

meteorological analyses is of particular interest to Lagrangian studies that potentially use these data to study PSC formation (Engel et al., 2013; Hoyle et al., 2013; Grooß et al., 2014; Di Liberto et al., 2015). Here we used the AIRS data to evaluate explicitly resolved temperature fluctuations due to gravity waves in the European Centre for Medium-Range Weather Forecast (ECMWF) operational analysis for a set of 21 large-amplitude gravity wave events in the polar winter seasons. With the spatial resolution and physical representation of the forecast models improving over time as well as the inclusion of new observations

within the assimilation procedures, it is important to know how realistically the analysis captures atmospheric gravity waves. As a second application, we performed a survey of joint AIRS gravity wave observations and PSC observations from the Michelson Interferometer for Passive Atmospheric Sounding (MIPAS) instrument (Fischer et al., 2008) aboard ESA's Envisat satellite. The survey covers the time period from 2003 to 2012, while Envisat MIPAS was in operation and provided nearly continuous monitoring of PSCs throughout the polar stratosphere. The main aim of this survey is to infer if the new AIRS

data set introduced here can help to better identify gravity wave-induced PSC formation events and to stimulate more detailed Lagrangian case studies in future work.

In Sect. 2 we provide a description of the AIRS instrument and the method used to extract gravity wave information from the measurements. The spatial and temporal patterns of wave activity in the lower stratosphere at mid and high latitudes as observed by AIRS are discussed in Sect. 3. The evaluation of explicitly resolved temperature fluctuations in the ECMWF operational

analysis is presented in Sect. 4. We discuss selected examples and the survey of gravity wave-induced PSC formation events based on AIRS and MIPAS data in Sect. 5. Finally, a summary and conclusions are given in Sect. 6.

## 2  AIRS observations of stratospheric gravity waves

AIRS (Aumann et al., 2003; Chahine et al., 2006) is one of six instruments aboard NASA's Aqua satellite. Aqua was launched in May 2002 into a nearly polar, low earth orbit at 705 km altitude, 100° inclination, and 100 min orbital period. The Aqua

orbit is sun-synchronous, with Equator crossings at 01:30 LT (descending nodes) and 13:30 LT (ascending nodes). Nearly global coverage is achieved during 14.4 orbits per day. AIRS measures infrared radiance spectra in a cross-track scanning geometry. Each scan consists of 90 footprints and covers 1780 km ground distance. The footprint size varies between $14 \times 14$ km$^2$ at nadir and $21 \times 42$ km$^2$ at the scan extremes. Adjacent scans are separated by 18 km along-track distance. The spectral measurements cover the $3.74 - 15.4\,\mu$m wavelength range in three bands. Brightness temperature measurements in the 4.3 and

15 $\mu$m wavebands of $CO_2$ are particularly suited to study stratospheric gravity waves (e. g., Alexander and Barnet, 2007; Hoffmann and Alexander, 2009; Gong et al., 2012; Hoffmann et al., 2014).

For this study, we identified a set of 21 AIRS channels in the 15 $\mu$m $CO_2$ waveband to study gravity wave activity in the polar lower stratosphere. Our channel selection differs from earlier work of Hoffmann et al. (2013, 2014), which used AIRS channels





in the 4.3 $\mu$m $CO_2$ waveband to obtain gravity wave information for the mid and upper stratosphere. The channels selected for this study are listed in Table 1, together with centroid frequencies and noise estimates (referred to as noise equivalent delta temperature, NeDT) at 250 K scene temperature. The radiance measurements of the selected channels are averaged to obtain a low-noise data product. Figure 1 shows the corresponding spectral mean temperature weighting functions for different

atmospheric conditions and for different viewing directions of AIRS. The weighting function for polar winter conditions at nadir shows maximum sensitivity to stratospheric temperatures around 23 km altitude. Its full-width at half-maximum is 15 km and extends over the altitude range from 17 to 32 km, which provides overlap with the altitude range typically covered by PSCs (Poole and Pitts, 1994; Spang et al., 2005; Pitts et al., 2009). Comparing the weighting function for nadir and the outermost scan angles we found that both have nearly the same width, but that the latter is shifted upward by about 2 km. This shift is

due to opacity growing more rapidly with altitude along slant ray paths. As stratospheric temperatures increase with altitude, measured radiances also increase with increasing scan angles, which is referred to as 'limb-brightening effect'. Although we are mainly interested in polar winter conditions, we also calculated weighting functions for polar summer conditions to assess the influence of atmospheric variability. In polar summer the temperature weighting functions are slightly broader and shift upward by about 2 km. This is due to stratospheric air density being higher in polar summer than in polar winter.

The temperature weighting functions can be used to calculate the amplitude response of the AIRS observations to gravity waves with different vertical wavelengths. Following the approach of Hoffmann and Alexander (2010) and Hoffmann et al. (2014), we convolve plane wave temperature perturbation profiles with known amplitude and vertical wavelength with the AIRS weighting functions. This provides simulated brightness temperature perturbations. Then the response is calculated as the ratio of the simulated brightness temperature perturbations and the true temperature amplitudes of the gravity waves. The

amplitude response curves for the AIRS 15 $\mu$m data set are shown in Fig. 1. The AIRS observations are limited to gravity waves with long vertical wavelengths due to the broad weighting functions of the nadir observation geometry. The amplitude response is about 10, 20, and 50% for vertical wavelengths of about 15, 20, and 37 km, respectively. For instance, assuming gravity waves with a vertical wavelength of 20 km, this means that the brightness temperature amplitudes are damped by a factor of 5 in the AIRS observations compared to the real temperature amplitudes of the gravity waves. For smaller vertical wavelengths the

attenuation will be stronger. As a consequence, brightness temperatures need to be measured with high precision to reliably detect the small perturbations caused by gravity waves. Reducing the measurement noise by spectrally averaging the radiance measurements of multiple AIRS channels helps to improve the response to gravity waves with short vertical wavelengths. Figure 1 shows that the amplitude response varies with respect to the atmospheric conditions, but that there are nearly no variations with respect to the AIRS scan angle.

Slowly-varying background signals need to be removed to extract gravity wave signals from the AIRS measurements. Background signals are caused by large-scale temperature gradients and planetary waves, but also by the limb-brightening effect. A standard detrending technique for AIRS is to remove the background defined by means of a polynomial fit to the brightness temperature measurements of each across-track scan (Wu, 2004; Eckermann et al., 2006; Alexander and Barnet, 2007; Hoffmann et al., 2014). Usually a 4th-order polynomial is applied for this purpose. However, we identified problems with this

approach in this study. Large-scale temperature gradients at the edge of the polar vortex can become rather large in the lower





stratosphere and are sometimes not completely captured by the 4th-order polynomial. Therefore we replaced the 4th-order polynomial by a 6th-order polynomial, noting that the higher-order fit is still well constrained by the measurements. The 6th-order polynomial fit reduces the amplitude response to gravity waves with long horizontal wavelengths, but it also effectively removes unwanted temperature residuals near the polar vortex edge. Following Hoffmann et al. (2014), we estimated amplitude

response levels of 50, 20, and 10% at 900, 1160, and 1350 km across-track wavelength, respectively (see Fig. 2). The limit for short horizontal wavelengths varies between 30 and 80 km, depending on the AIRS scan angle.

The analysis of AIRS gravity wave observations requires careful characterization of measurement noise. Accurate noise estimates are particularly important if brightness temperature variances are calculated for long time periods or large regions because gravity wave signals tend to average out in this case. We applied the approach of Hoffmann et al. (2014) to estimate

the noise of the AIRS 15 µm brightness temperature data set directly from the measurements. Figure 3 shows that the noise estimates vary with scene temperature. The noise typically ranges from 0.109 K at 250 K scene temperature to 0.201 K at 195 K scene temperature. The highest noise is found in polar winter conditions, where scene temperatures are coldest and measured radiance signals are lowest. In contrast, lowest noise occurs in polar summer conditions. Note that the noise of the spectrally averaged data set is about 4 – 6 times lower than the nominal noise levels of the individual AIRS channels (Table 1). This

provides a substantial improvement in terms of sensitivity to gravity waves with short vertical wavelengths. Note also that the dependence of noise on the scene temperature is well characterized by a Planck scaling curve (Fig. 3). Following Hoffmann et al. (2014), this scaling curve is used in our analyses to subtract noise estimates from brightness temperature variances, so that only gravity wave signals remain. The variances are then referred to as 'noise corrected'. The correction has an accuracy of about 2%, which is determined by the uncertainty of fit of the Planck function.

As an example, Fig. 4 shows AIRS measurements at both 8.1 and 15 µm wavelength on 24 August 2004. This particular day is characterized by rather strong gravity wave activity at southern mid and high latitudes. The 15 µm brightness temperature map shows large-scale temperature gradients and planetary wave activity associated with the polar vortex. The limb-brightening effect is also visible, yielding increased brightness temperatures toward the scan edges. Gravity waves are hardly visible in the raw 15 µm brightness temperature map, but are isolated by means of the detrending method. The 15 µm brightness temperature

perturbation map reveals gravity wave activity over the Southern Andes as well as the Transantarctic Mountains and Mac Robertson Land. Previous climatological studies using AIRS data showed that these regions are in fact hotspots of stratospheric gravity wave activity (Gong et al., 2012; Hoffmann et al., 2013, 2016). AIRS measurements in the spectral window region at 8.1 µm can be used to detect storm systems (Aumann et al., 2006; Hoffmann and Alexander, 2010). The 8.1 µm brightness temperature map indicates several intense storm systems over the southern oceans, with cloud top temperatures as low as

210 – 230 K. However, the 15 µm brightness temperature map shows that these storm systems do not cause any direct radiance signals in the stratospheric channels. Note that this only applies for mid and high latitudes. At low latitudes the tropopause is higher and clouds can reach the altitude range covered by the selected channels. Our data should not be used at low latitudes.

Figure 4 also shows detrended and noise-corrected 15 µm brightness temperature variances on a 4° × 2° longitude-latitude grid calculated from the brightness temperature perturbations. Although AIRS already provides about 80% global coverage

during 12 h time intervals, we decided to combine data over a 24 h time window to further close data gaps and homogenize



temporal coverage. The $4° \times 2°$ grid boxes typically contain $160 - 520$ footprints at mid and high latitudes, which allows us to calculate variances with quite low sampling errors. Although AIRS provides measurement coverage up to the poles, any data beyond $\pm 85°$ latitude are excluded in this analysis. This is simply because the sampling coverage on a longitude-latitude grid decreases rapidly towards the poles. This could be mitigated by changing to another sampling grid, but we did not find this

to be important because gravity wave activity beyond $\pm 85°$ latitude usually is rather low in the AIRS observations. Figure 4 shows that the variance map captures the individual gravity wave events found in the perturbation map. It also reveals that the noise correction very effectively suppresses noise signals in the variances. The map plot suggests that gravity wave signals can be reliably detected for variances as low as $5 \times 10^{-3}\,\mathrm{K}^2$. Variances below this threshold can be affected by sampling errors. Note that based on an estimated 2% uncertainty of the noise correction, the corresponding uncertainty of the variances is in

the range of $2.4 \times 10^{-4}\,\mathrm{K}^2$ at 250 K scene temperature and $7.8 \times 10^{-4}\,\mathrm{K}^2$ at 195 K scene temperature, which is well below the sampling error. We calculated the gridded $15\,\mu\mathrm{m}$ brightness temperature variances on the $4° \times 2°$ grid on a daily basis for the entire $2003 - 2012$ time period for further analyses.

## 3 Temporal and spatial patterns of gravity wave activity

In this section we discuss the spatial and temporal patterns of the gravity wave activity in the lower stratosphere at mid and
high latitudes as observed by AIRS during the years 2003 to 2012. First, we focus on daily variations. Figure 5 shows time series of detrended and noise-corrected $15\,\mu\mathrm{m}$ brightness temperature variances integrated over $85°\mathrm{S} - 55°\mathrm{S}$ and $55°\mathrm{N} - 85°\mathrm{N}$, respectively. As gravity wave activity typically occurs rather locally (cf. Fig. 4), the variances calculated for such large regions decrease to low levels. Maxima in different years range from $4.6 \times 10^{-3}$ to $1.0 \times 10^{-2}\,\mathrm{K}^2$ in the northern hemisphere and $7.2 \times 10^{-3}$ to $2.1 \times 10^{-2}\,\mathrm{K}^2$ in the southern hemisphere. However, note that these values are still considerably larger than

the uncertainties of the noise correction. Figure 5 reveals that the daily variations of the gravity wave activity are substantial. Nevertheless, after detrending the daily time series with a 30-day running mean, the mean autocorrelation for a time lag of one day is 0.63 in the northern hemisphere and 0.57 in the southern hemisphere. The interannual variation of the autocorrelation is $\pm 0.16$ in the northern hemisphere and $\pm 0.14$ in the southern hemisphere. Despite the variability of the time series, this indicates that there is some persistence of gravity wave activity, which we attribute to the persistence of both the gravity wave

sources and the tropospheric and stratospheric background winds.

The time series for both the northern and southern hemisphere in Fig. 5 show a distinct seasonal cycle of gravity wave activity with maxima in the winter months. Gravity wave activity is typically stronger and lasts longer in the southern hemisphere, whereas the interannual variability and the intraseasonal variations are larger in the northern hemisphere. A statistical analysis of the seasonal cycle of gravity wave activity is presented in Fig. 6. We calculated the 10, 25, 50, 75, and 90% quantiles of

the $15\,\mu\mathrm{m}$ brightness temperature variance distributions for 30-day time windows. The analysis is performed with the time windows being centered around the beginning or the middle of each month. Based on the 10% and 90% quantiles, we found that the seasonal cycle of gravity wave activity in the northern hemisphere typically begins in November and ends sometime in February to April. The cycle lasts about $2 - 6$ months. Maximum activity occurs during January. In the southern hemisphere the





cycle begins in April to June and ends in November or December. The cycle lasts about 5 – 9 months. Maximum activity occurs during August. For comparison, we also analyzed the seasonal cycle for two orographic hotspots of gravity wave activity, the Scandinavian Mountains and the Antarctic Peninsula (Fig. 6). Local variances at the hotspots are much larger than the polar averages. For instance, the local maximum of the 90% quantile exceeds the polar average by a factor of 9 for the Antarctic Peninsula and by a factor of 4 for the Scandinavian Mountains. The variance distributions for the hotspots show positive skewness, with a long tail towards large variances, as can be seen by comparing the ranges between the 10% and 90% quantiles and the median. This indicates that few, but strong mountain wave events contribute substantially to gravity wave activity at these hotspots.

In order to analyze the spatial patterns and to identify source regions of gravity wave activity we calculated monthly variances on the $4° \times 2°$ longitude-latitude grid directly from the daily data. In Figs. 7 and 8 we also show maps of terrain height standard deviations calculated from a 2-minute gridded global relief data set (ETOPO2v2; National Geophysical Data Center, 2006) to indicate possible orographic sources of gravity wave activity. The AIRS maps indicate that gravity wave activity increases significantly at a number of orographic hotspots. In the northern hemisphere this includes (from west to east) the Arctic Archipelago, Labrador, Greenland, Iceland, the United Kingdom, the Alps, the Scandinavian Mountains, the Ural Mountains, the Altai Mountains, the Central Siberian Plateau, and the East Siberian Plateau. Gravity wave activity over the Rocky Mountains is largely absent, which is due to relatively low stratospheric background winds over this mountain range (Hoffmann et al., 2013, 2014). In the southern hemisphere we found increased gravity wave activity in a latitude band around $70°S – 50°S$ during the winter months, which is attributed mostly to jet and storm sources (Sato et al., 2012; Hendricks et al., 2014; Hindley et al., 2015). Orographic hotspots can be identified at the southern Andes, the Antarctic Peninsula, South Georgia, Kerguelen Islands, and the Transantarctic Mountains. The spatial patterns of gravity wave activity in the lower stratosphere found here agree well with those identified by Gong et al. (2012) and Hoffmann et al. (2013, 2014, 2016) for the mid and upper stratosphere.

Strong background winds in the stratosphere tend to go along with unidirectional and increasing winds with height (e. g., Orr et al., 2010). In this case wave refraction will result in long vertical wavelengths (e. g., Wu and Eckermann, 2008), which are best visible to AIRS. On the contrary, weak background winds are associated with gravity waves with short vertical wavelengths, which are generally not detectable by AIRS. This combined effect of the stratospheric background winds and the vertical wavelength sensitivity of AIRS is referred to as an 'observational filter' (Alexander and Barnet, 2007; Hoffmann et al., 2014, 2016). Figures 7 and 8 also shows contours of monthly mean zonal winds at 30 hPa (about 25 km altitude) from the ERA-Interim reanalysis (Dee et al., 2011). The 30 hPa level is close to the altitude range where the AIRS observations provide maximum sensitivity to stratospheric temperatures. Figures 7 and 8 show that the 30 hPa zonal winds have a significant influence on the observability of gravity waves with AIRS. For both the northern and southern hemisphere we found that strong westerlies are a prerequisite for the observation of gravity wave activity with AIRS. The comparison suggests that zonal winds need to exceed levels of about $10 – 20\,\mathrm{m\,s^{-1}}$ before gravity wave activity becomes visible in the AIRS data. This shows that the observational filter needs to be taken into account in the analysis of AIRS gravity wave observations.



## 4    Evaluation of temperature fluctuations in the ECMWF operational analysis

In this study we used the new AIRS data set to evaluate explicitly resolved temperature fluctuations due to gravity waves in the ECMWF operational analysis. During the time period considered here (2003 – 2012) the horizontal resolution of the ECMWF operational analysis was increased from T511 (39 km effective resolution) to T1279 (16 km), the number of model

levels was increased from 60 to 91, and the model top was raised from 0.1 to 0.01 hPa. The typical vertical resolution is about 0.5 – 1.1 km at 15 – 30 km altitude. We here considered analysis data at 0 and 12 UTC and forecast data at 3, 6, 9, 15, 18, and 21 UTC. AIRS radiance data have been used for operational forecasting since October 2003 (McNally et al., 2006; Dee et al., 2011). Observations of up to 210 channels are used in the assimilation procedure, of which 10 are also considered in our AIRS data product. The comparisons of AIRS and ECMWF data presented here are therefore not strictly based on independent data

and should be considered as an 'evaluation' rather than a 'validation'. However, the assimilation of AIRS data at ECMWF is restricted to cloud-free scenes and an additional random thinning operation is applied to ensure a minimum horizontal spacing of 120 km between the footprints. This poses a Nyquist limit of 240 km on horizontal wavelengths regarding the direct assimilation of small-scale gravity waves from AIRS observations into the analysis. Wave patterns with shorter horizontal wavelengths are most likely generated by the model itself.

In our comparison we took the effects of radiative transfer and the AIRS satellite observation geometry into account. First, we extracted temperature profiles from the ECMWF data at the locations of individual satellite footprints by means of 3-D linear interpolation. Next, we convolved the temperature profiles with the weighting function of the selected channels. This gave us simulated brightness temperature measurements on the AIRS measurement grid. Gravity wave signals were then extracted by means of the same detrending procedure as used for the real AIRS data. The main difficulty in this comparison

is related to the coarse time resolution (3-hourly) at which the ECMWF data are provided. We tested different interpolation schemes (nearest neighbor, linear, and cubic), but found that temporal interpolation often does not provide physically meaningful solutions for gravity waves that may propagate through the troposphere and stratosphere within just a few hours. Therefore we decided to avoid time interpolation altogether and to perform the analysis directly with the meteorological data at the 3 h synoptic time steps. Daily gravity wave variances were calculated using the data of all synoptic time steps on each day. This

approach introduces some uncertainty in the analysis, but it was found to still provide the best overall agreement with real AIRS observations.

As an example, Fig. 9 shows a qualitative comparison of 15 µm brightness temperature perturbations from real AIRS measurements and corresponding simulated measurements based on the ECMWF operational analysis data. Real measurements shown here were obtained on 12 December 2003, 12:00 to 24:00 UTC in the northern hemisphere and 22 July 2011, 00:00 to

12:00 UTC in the southern hemisphere. The simulated measurements apply to 18:00 and 06:00 UTC synoptic time on these days, respectively. The AIRS measurements in the northern hemisphere show mountain waves near the east coast of Greenland as well as the Scandinavian Mountains. There is also a band of wave activity around 60 – 75°N extending over large parts of Asia. The wave activity in this region can be attributed to orography, but also to jet and storm sources. The example for the southern hemisphere shows strong mountain waves at the southern Andes. The measurements also reveal non-orographic wave



activity related to jet and storm sources possibly mixed with orographic waves near coastal East Antarctica at $50 - 70°$S and $60°$E $- 120°$W. The qualitative comparison shows that both orographic and non-orographic waves are captured by the ECMWF operational analysis to a great extent, i. e., there is remarkably good agreement between the regions where gravity waves were represented in the analysis and where they actually occurred. Despite the time differences, even the details of the wave pat-

terns look similar, in particular for the southern hemisphere example in July 2011. The improved horizontal resolution of the ECMWF operational analysis is the likely reason why individual patterns are better captured in the southern hemisphere example for the year 2011 compared to the northern hemisphere example for the year 2003. However, we note that wave amplitudes from the simulations are lower than the wave amplitudes from the AIRS measurements in both examples.

To quantify the systematic differences between the wave amplitudes in the AIRS observations and the meteorological anal-

yses we analyzed seasonal peak events. We selected the day with the largest gravity wave variance from the AIRS time series (Fig. 5) for each winter season in the northern and southern hemisphere. Figure 10 shows a bar chart of $15\,\mu$m brightness temperature standard deviations for these peak events, calculated from the detrended and noise-corrected $15\,\mu$m brightness temperature variances shown in Fig. 5. Here we present standard deviations rather than variances, because those can be related more directly to gravity wave amplitudes. Figure 10 shows that the simulated wave amplitudes almost always underestimate

observed wave amplitudes. On average, the ECMWF operational analysis reproduces $(32 \pm 13)\%$ of the observed standard deviations in the northern hemisphere and $(47 \pm 22)\%$ for the southern hemisphere. Commonly, gravity wave amplitudes are better reproduced in the southern hemisphere than in the northern hemisphere. This difference between the hemispheres is related to differences of the typical gravity wave spectra in both hemispheres. For the 21 peak events analyzed here, a 2-D spectral analysis of the AIRS data using the S-Transform method (Stockwell et al., 1996; Alexander and Barnet, 2007) yielded

mean horizontal wavelengths of $(116 \pm 52)\,$km in the northern hemisphere and $(133 \pm 72)\,$km in the southern hemisphere. These differences might be due to the gravity wave sources such as orography, convection, and jets as well as differences of the background flow between the hemispheres. The model produces shorter horizontal wavelength waves in the northern hemisphere, however, the wavelengths were overall longer than AIRS, and we found stronger attenuation of the wave amplitudes in the model. We attribute the interannual variability of the scaling factors of the wave amplitudes between the AIRS and

ECMWF data to atmospheric variability, but also to improvements in the spatial resolution, the forecast model, and the data assimilation system of the operational analysis over time.

## 5   A survey of gravity wave-induced PSC formation events

In this section we present a survey of joint AIRS gravity wave and Envisat MIPAS PSC observations during the years 2003 to 2012. Like Aqua, Envisat operated in a nearly polar, low earth orbit (790 km altitude, $98°$ inclination, 101 min orbital pe-

riod) with Equator crossings at 10:00 LT and 22:00 LT. Envisat MIPAS (Fischer et al., 2008) measured $4.15 - 14.6\,\mu$m limb emission spectra of atmospheric constituents from the mid troposphere to the mesosphere at high spectral resolution. Nominal measurements in the 'full resolution' phase ($0.025\,\mathrm{cm}^{-1}$ spectral resolution) in $2002 - 2004$ provided 550 km horizontal sampling and 3 km vertical sampling in the lower stratosphere. This was improved to 410 km and $1.5 - 2$ km in the 'optimized




resolution' phase ($0.0625\,\mathrm{cm}^{-1}$) in $2005-2012$. The instantaneous field of view of MIPAS covers $3-4\,\mathrm{km}$ (elevation) $\times\,30\,\mathrm{km}$ (azimuth). Several studies demonstrated that MIPAS was capable of detecting PSCs based on specific spectral signatures in the mid infrared (Spang et al., 2004; Höpfner et al., 2006a, b; Spang et al., 2012, 2014). A particular advantage of MIPAS was its sensitivity to low particle concentrations due to the long integration paths of the limb observation geometry. Here we

make use of a new MIPAS PSC data product introduced by Spang et al. (2016), which uses radiance measurements at 7.1, 8.2, 10.5, 12.0, and $12.7\,\mu\mathrm{m}$ to detect PSCs and to classify between different particle types. The new method has been developed and tested with a database of radiative transfer model calculations of realistic PSC particle size distributions, geometries, and composition. The detection and classification results were compared with space-borne lidar observations. Spang et al. (2016) showed that ice particles are classified most accurately with the new method.

Our survey of gravity wave-induced PSC formation events is based on daily maps of detrended and noise-corrected AIRS $15\,\mu\mathrm{m}$ brightness temperature variances and MIPAS PSC detections. Selected examples of these maps are shown in Fig. 11. MIPAS PSC detections are shown for the altitude range of about $18-22\,\mathrm{km}$ ($450-550\,\mathrm{K}$ potential temperature), which coincides with the range of polar vortex temperature minima (Randel et al., 2004) and PSC occurrence frequency maxima (Poole and Pitts, 1994; Spang et al., 2005; Pitts et al., 2009) during the course of the polar winter. To help link the PSC observations

to gravity wave activity, further information was added to the maps, following Spang et al. (2016). PSC existence temperatures were estimated following Hanson and Mauersberger (1988) and Marti and Mauersberger (1993), based on ERA-Interim pressure and temperature data at the 500 K isentropic level as well as typical stratospheric values for $HNO_3$ (9 ppbv) and $H_2O$ (4 ppmv). PSC detections associated with synoptic-scale temperatures significantly larger than the specific existence temperature are candidates for gravity wave-induced formation events. Furthermore, the maps include contours of the Montgomery

streamfunction (calculated from ERA-Interim data), representing streamlines of the geostrophic wind at the 500 K isentropic level. The streamfunctions illustrate the flow of air masses on short time scales and can be used to infer if the MIPAS PSC detections occurred downstream of AIRS gravity wave observations. This helps to overcome difficulties related to the miss-times between the Aqua and Envisat overpasses and the synoptic time steps of the meteorological data.

Figure 11 shows two examples of gravity wave-induced PSC formation events in the northern hemisphere. On 25 January

2007, MIPAS detected ice PSCs over Scandinavia as well as Germany and Poland at synoptic-scale temperatures up to $6-9\,\mathrm{K}$ above $T_{ice}$. These detections are located downstream of Greenland, where strong gravity wave activity was present at the east and north coast according to the AIRS observations. Weak gravity wave activity is also visible over the Scandinavian Mountains. On 7 January 2011, MIPAS detected ice, STS, and NAT PSC over the Arctic Archipelago at synoptic-scale temperatures about 3 K above $T_{NAT}$. Although the Arctic Archipelago was found to be a gravity wave hotspot in the northern hemisphere

(Sect. 3), no indication for gravity wave activity is seen in this region on this particular day in the AIRS data. However, gravity wave activity occurred upstream over the Beaufort Sea, which may suggest that non-orographic gravity waves could play a role in this case.

Figure 11 also presents three examples of gravity wave-induced PSC formation in the southern hemisphere. The measurements on 8 July 2007 show a rather typical example of the formation of ice PSCs due to mountain waves at the Antarctic

Peninsula. The ice PSCs are formed at synoptic-scale temperatures up to $4-10\,\mathrm{K}$ above $T_{ice}$. NAT particles are observed fur-





ther downstream of the Antarctic Peninsula, which is consistent with the ice particles serving as condensation nuclei for NAT formation (Carslaw et al., 1998a; Höpfner et al., 2006b; Eckermann et al., 2009). Although mountain wave activity is most frequent at the Antarctic Peninsula (Hoffmann et al., 2016), other orographic features in Antarctica can also play a role in PSC formation. The AIRS and MIPAS observations on 25 July 2008 and 29 August 2008 show cases of mountain wave-induced formation of ice PSCs over the Transantarctic Mountains and near Enderby Land and Mac Robertson Land, respectively. The case on 29 August 2008 is particularly interesting, because it shows that AIRS provides important complementary information on gravity wave activity. Oscillations of the $T_{ice}$ contour line at the western longitudes of Antarctica indicate that gravity waves are explicitly resolved in the ERA-Interim reanalysis in this region. However, the formation of ice PSCs near Enderby Land and Mac Robertson Land is associated with gravity wave activity observed by AIRS, which is not captured by the ERA-Interim data at all.

In Figure 11 we displayed 15 µm brightness temperature variances up to a level of 0.075 K$^2$. A relatively low cut-off was selected here to make also weak wave signals visible. Following the approach of Alexander and Grimsdell (2013), we performed a cross check for the orographic wave events to confirm that these are indeed cases with substantial true gravity wave amplitudes. We computed the vertical wavelength $\lambda_z$ from the gravity wave dispersion relation for stationary waves, assuming that the ground-based frequency and phase speed are zero,

$$\lambda_z = 2\pi \left[ \left( \frac{N}{u} \right)^2 - \left( \frac{2\pi}{\lambda_x} \right)^2 \right]^{-0.5} . \tag{1}$$

Here $N \approx 0.02 \, \text{s}^{-1}$ refers to the stratospheric buoyancy frequency. The background wind speed $u$ was taken from the ERA-Interim reanalysis at 30 hPa. The horizontal wavelength $\lambda_x$ was determined by means of spectral analysis of the AIRS data. The vertical wavelength estimate $\lambda_z$ was used to determine the amplitude response $R$ from the data shown in Fig. 1. Then $R$ was used to estimate the true gravity wave amplitude $T'$ from the measured brightness temperature amplitude $BT'$, according to $T' = BT' R^{-1}$. The results of the calculations for the four orographic wave events in Fig. 11 are summarized in Table 4. Based on background wind speeds of about $55 - 65 \, \text{m s}^{-1}$ at 30 hPa and horizontal wavelengths of about $95 - 130 \, \text{km}$, we found vertical wavelengths of about $17 - 21 \, \text{km}$. The corresponding amplitude response is about $15 - 22\%$. Brightness temperature amplitudes of about $1.3 - 2.6 \, \text{K}$ scale to true gravity wave amplitudes of about $7 - 12 \, \text{K}$. Such large true amplitudes are indeed consistent with observations of ice PSCs well above the frost point.

Based on the daily maps we performed a survey of all AIRS and MIPAS observations in the northern and southern polar winter seasons during the years 2003 to 2012. In this survey we identified 48 events of gravity wave-induced PSC formation (Tables 2 and 3), with 29 events being located in the southern hemisphere and 19 events being located in the northern hemisphere. This first survey suggests that formation events appear more frequently in the southern hemisphere than in the northern hemisphere. This is not consistent with studies showing that there are larger proportions of PSCs due to orographic gravity waves in the Arctic than in the Antarctic (Kohma and Sato, 2011; Alexander et al., 2013). However, our findings might be biased due to the more stable atmospheric conditions in the southern hemisphere that ease the identification of events by means of visual inspection of the AIRS and MIPAS maps and stronger winds in the southern hemisphere that make waves more visible in AIRS data. In the southern hemisphere, most events are found near the Antarctic Peninsula (20 events), followed by the





Transantarctic Mountains (4), and other places of coastal Antarctica. In the northern hemisphere, most events are found near Greenland (6), Iceland (4), and the Scandinavian Mountains (3). The dates of the events and possible source regions of gravity wave activity are listed in Tables 2 and 3. The tables provide suggestions for possible case studies that could be exploited in future work, e. g., by means of Lagrangian trajectory analyses. A large number of events identified here is promising and

supports the conclusion that gravity wave-induced PSC formation plays an important role in polar ozone chemistry. It shows that the AIRS record of gravity wave activity in the lower stratosphere is a valuable complementary resource that can be used to study this process in more detail.

# 6   Summary and conclusions

Here we introduced a new long-term satellite record of gravity wave activity in the lower stratosphere at mid and high latitudes.

The data set was compiled to support studies on the influence of small-scale temperature fluctuations due to gravity waves on PSC formation. The record was derived from AIRS/Aqua observations between January 2003 and December 2012. Gravity wave activity is measured in terms of detrended and noise-corrected $15\,\mu$m brightness temperature variances on a daily basis. We discussed the characteristics of the AIRS channels selected for this analysis in terms of vertical coverage, wavelength sensitivity, and measurement noise. The analysis of temporal patterns of the AIRS gravity wave observations revealed a strong

seasonal cycle with wintertime maxima at both northern and southern hemisphere mid and high latitudes. The analysis of spatial patterns showed that the wave activity is mostly related to orographic, jet, and storm sources. The observed patterns of wave activity agree well with those reported in other climatological studies of global gravity wave activity using AIRS observations (Gong et al., 2012; Hoffmann et al., 2013, 2014). The observed gravity wave activity is closely correlated with zonal winds at the $30\,$hPa level, which we attribute to the AIRS observational filter. Typically, a background wind of at least $10-20\,$m s$^{-1}$ is

required to foster the propagation of gravity waves with long vertical wavelengths into the lower stratosphere, which are best visible to AIRS. Small vertical wavelengths are associated with significant attenuation of the observed brightness temperature perturbations. This attenuation needs to be taken into account if the AIRS data are compared with other measurements or model results. This can be achieved by convoluting temperature perturbation profiles from the comparative data with the AIRS weighting functions. Alternatively, the brightness temperature perturbations can be scaled up to estimate true wave amplitudes,

but this requires information on vertical wavelength for any particular case.

We discussed two applications of the new AIRS long-term record of gravity wave activity. The first application is an evaluation of explicitly resolved gravity waves in the ECMWF meteorological data products. We found that observed and simulated gravity wave patterns agree well in extent and shape. However, the amplitudes of the short horizontal and long vertical wavelength gravity waves that are best visible to AIRS are typically underestimated by a factor of $2-3$ by the ECMWF operational

analysis. This is in line with results of Schroeder et al. (2009) and Jewtoukoff et al. (2015), who validated temperature fluctuations and gravity wave momentum fluxes in the ECMWF operational analysis with Infrared limb sounding measurements and superpressure balloon observations, respectively. Both studies attributed the underestimation of gravity wave amplitudes and momentum fluxes to the spatial truncation of the ECMWF model. The second application presented here is a survey of gravity





wave-induced PSC formation events based on joint AIRS and MIPAS observations. Envisat MIPAS observations during the years 2003 to 2012 revealed nearly 50 events of PSC detections at synoptic-scale temperatures well above the PSC existence and formation thresholds. In many cases we found that the detections occurred downstream of source regions with gravity wave activity as revealed by AIRS observations. The large number of events found in this survey confirms that gravity wave-induced PSC formation is indeed an important process in polar ozone chemistry. The events found here can be explored in more detail by means of Lagrangian trajectory analyses in future work.

## 7 Data availability

AIRS data are distributed by the NASA Goddard Earth Sciences Data Information and Services Center (AIRS Science Team and Chahine, 2007). Envisat MIPAS Level-1B data are distributed by the European Space Agency. The ERA-Interim reanalysis (Dee et al., 2011) and the operational analysis were obtained from the European Centre for Medium-Range Weather Forecasts. The ETOPO2v2 data set was obtained from the US Department of Commerce, National Oceanic and Atmospheric Administration, National Geophysical Data Center (National Geophysical Data Center, 2006). Interested scientists can obtain access to the complete AIRS gravity wave data set introduced in this paper by contacting the leading author.

*Acknowledgements.* MJA and LAH acknowledge the NASA Goddard Space Flight Center for support (grant #NNX14AO76G).





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



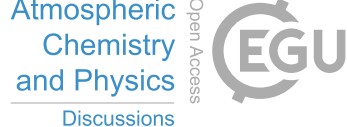

**Table 1.** AIRS Radiance Channels of the 15 $\mu$m Brightness Temperature Data Product

| Channel Number | Wave-number [cm$^{-1}$] | NEdT at 250 K [K] | Channel Number | Wave-number [cm$^{-1}$] | NEdT at 250 K [K] |
|---|---|---|---|---|---|
| 5 | 650.6 | 0.60 | 101 | 674.4 | 0.42 |
| 11 | 652.0 | 0.67 | 102 | 674.7 | 0.44 |
| 17 | 653.5 | 0.61 | 107 | 676.0 | 0.45 |
| 23 | 654.9 | 0.61 | 108 | 676.2 | 0.42 |
| 30 | 656.6 | 0.54 | 113 | 677.5 | 0.39 |
| 36 | 658.1 | 0.51 | 114 | 677.8 | 0.43 |
| 42 | 659.6 | 0.47 | 119 | 679.1 | 0.42 |
| 56 | 663.0 | 0.46 | 120 | 679.4 | 0.45 |
| 84 | 670.1 | 0.49 | 125 | 680.7 | 0.41 |
| 89 | 671.3 | 0.39 | 126 | 680.9 | 0.41 |
| 95 | 672.9 | 0.43 | | | |

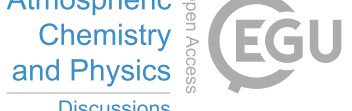



**Table 2.** Northern Hemisphere Gravity Wave-induced PSC Formation Events

| Date | Source Region | Date | Source Region |
|------|---------------|------|---------------|
| 2003-02-10 | Ural Mountains | 2008-01-01 | Iceland |
| 2005-01-27 | Greenland | 2008-01-28 | Iceland |
| 2005-02-16 | Greenland | 2008-02-10 | Iceland |
| 2005-02-18 | Greenland | 2008-12-24 | Iceland |
| 2006-01-19 | United Kingdom | 2010-01-03 | Greenland |
| 2006-12-26 | Scandinavian Mountains | 2010-12-07 | Greenland |
| 2007-01-25 | Greenland | 2011-01-06 | non-orographic |
| 2007-02-13 | Labrador | 2011-01-29 | East Siberian Plateau |
| 2007-12-21 | Scandinavian Mountains | 2011-12-27 | Scandinavian Mountains |
|  |  | 2011-12-29 | United Kingdom |





**Table 3.** Southern Hemisphere Gravity Wave-induced PSC Formation Events

| Date | Source Region | Date | Source Region |
|---|---|---|---|
| 2003-06-11 | Antarctic Peninsula | 2009-06-02 | Mac Robertson Land |
| 2003-06-30 | Antarctic Peninsula | 2009-06-27 | Antarctic Peninsula |
| 2003-08-23 | Transantarctic Mountains | 2009-07-19 | Antarctic Peninsula |
| 2003-09-10 | Antarctic Peninsula | 2009-07-24 | non-orographic |
| 2003-09-16 | Antarctic Peninsula | 2009-09-10 | Transantarctic Mountains |
| 2005-06-08 | Marie Byrd Land | 2010-06-11 | Antarctic Peninsula |
| 2005-06-14 | Antarctic Peninsula | 2010-07-21 | Antarctic Peninsula |
| 2006-06-28 | Antarctic Peninsula | 2010-07-30 | Marie Byrd Land |
| 2007-07-08 | Antarctic Peninsula | 2010-08-16 | Antarctic Peninsula |
| 2007-08-04 | Transantarctic Mountains | 2011-06-24 | Enderby Land |
| 2007-08-16 | Antarctic Peninsula | 2011-06-26 | Antarctic Peninsula |
| 2007-09-10 | Antarctic Peninsula | 2011-08-05 | Antarctic Peninsula |
| 2008-05-29 | Antarctic Peninsula | | |
| 2008-07-25 | Transantarctic Mountains | | |
| 2008-08-29 | Mac Robertson Land | | |
| 2008-09-12 | Antarctic Peninsula | | |
| 2008-09-22 | Antarctic Peninsula | | |





**Table 4.** Estimation of Gravity Wave Amplitudes for the Orographic Events Shown in Fig. 11

| Event Date | $u$ | $\lambda_x$ | $\lambda_z$ | $R$ | $BT'$ | $T'$ |
| --- | --- | --- | --- | --- | --- | --- |
| | [m s$^{-1}$] | [km] | [km] | [%] | [K] | [K] |
| 2007-01-25 | 65 | 113 | 20.7 | 21.6 | 1.8 | 8.3 |
| 2007-07-08 | 55 | 111 | 17.5 | 15.4 | 1.4 | 9.1 |
| 2008-07-25 | 65 | 128 | 20.7 | 21.6 | 2.6 | 12.0 |
| 2008-08-29 | 60 | 95 | 19.2 | 18.5 | 1.3 | 7.0 |



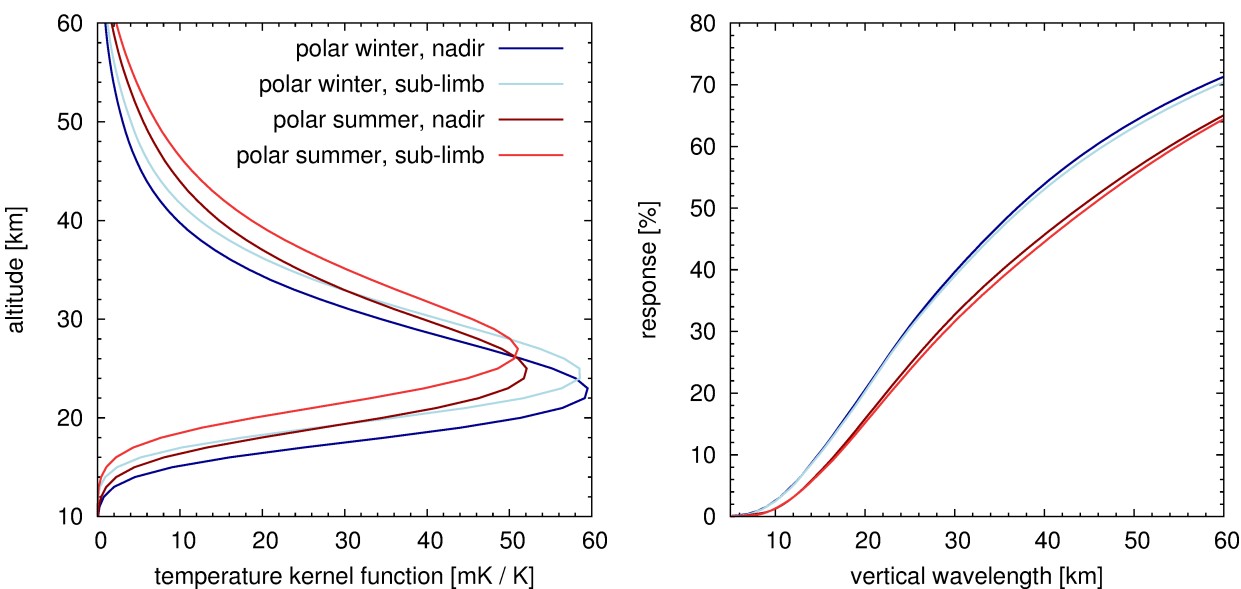

**Figure 1.** Temperature weighting functions (left) and amplitude response curves (right) for the AIRS 15 $\mu$m brightness temperature data set. Radiative transfer calculations have been performed for polar summer and polar winter conditions, the nadir direction and the outermost scan angles (referred to as 'sub-limb'), and a 1 km altitude grid.

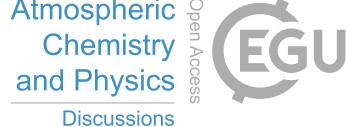



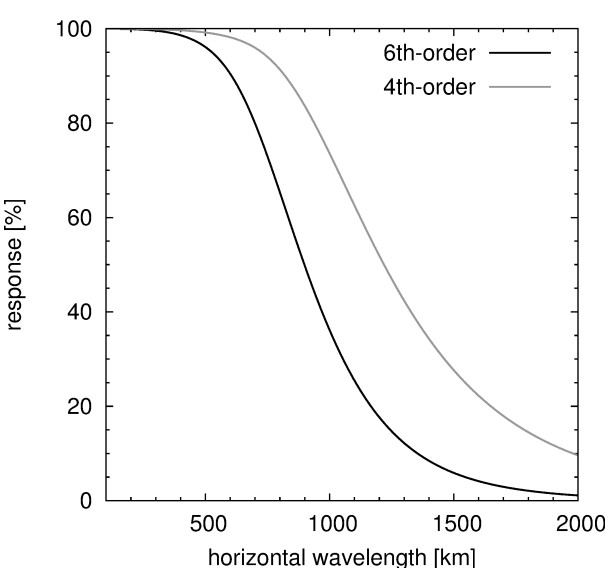

**Figure 2.** Amplitude response of the AIRS 15 $\mu$m brightness temperature data set for gravity waves with different across-track wavelengths. The plot shows the response using a 4th- or 6th-order polynomial fit to estimate background signals.





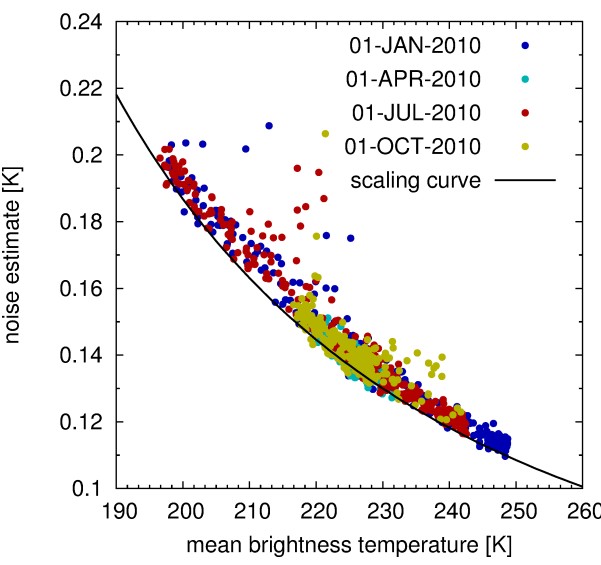

**Figure 3.** Noise estimates for the AIRS 15 $\mu$m brightness temperature data set versus mean background temperature. Individual noise estimates (colored dots) have been obtained from radiance measurements within globally distributed boxes of 90×90 satellite footprints on different days. Few outliers with high noise are due to cases with small-scale waves being miss-interpreted as noise. The Planck scaling curve (black curve) is defined by a NEdT of 0.109 K at 250 K scene temperature.



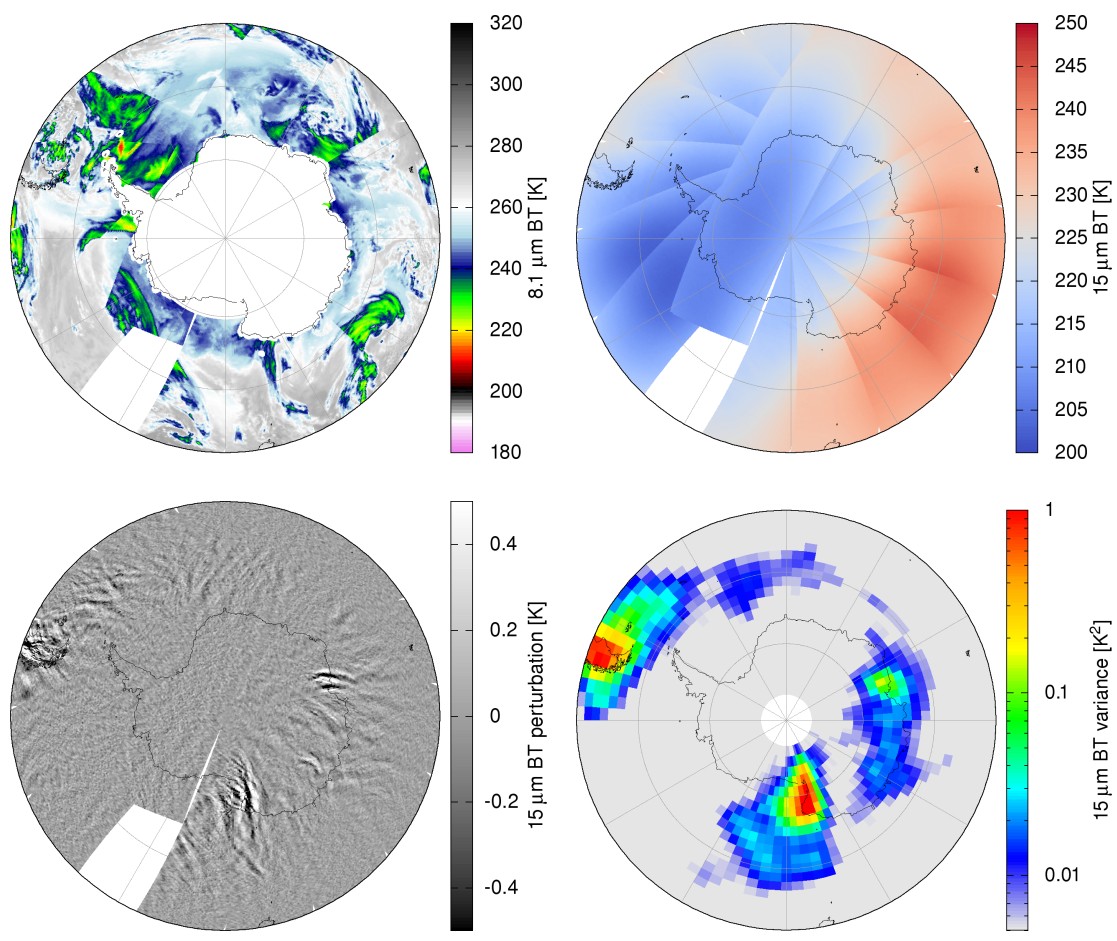

**Figure 4.** AIRS measurements of 8.1 $\mu$m (top, left) and 15 $\mu$m (top, right) brightness temperatures (BTs) on 24 August 2004, 12 – 24 UTC. The 8.1 $\mu$m BT observations show high clouds and storm systems. The 15 $\mu$m BT perturbation map (bottom, left) provides information on gravity waves in the lower stratosphere. BT variances on a 4° × 2° horizontal grid (bottom, right) are calculated by combining measurements from 00 – 12 UTC (not shown) and 12 – 24 UTC on the given day to close data gaps and to homogenize temporal coverage.



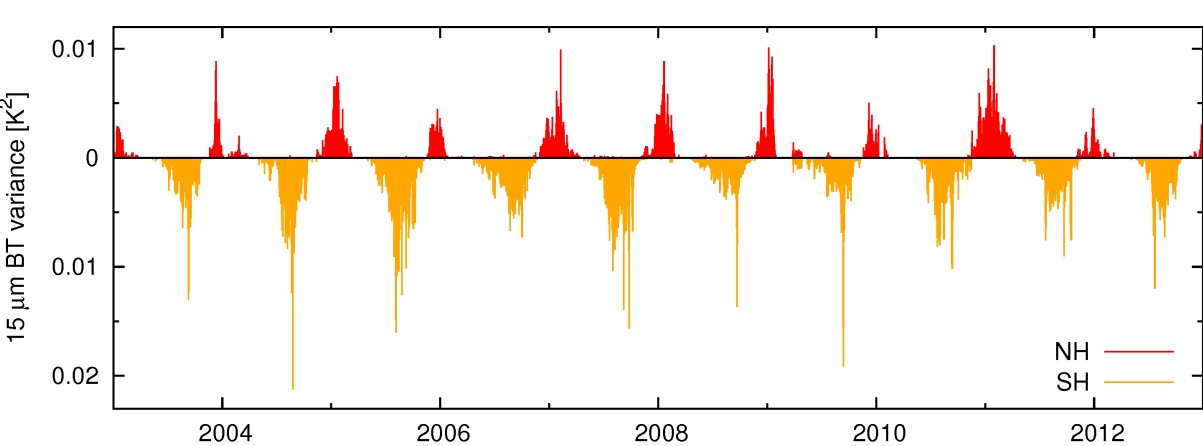

**Figure 5.** Time series of daily $15\,\mu$m brightness temperature variances from AIRS measurements at northern hemisphere (NH) and southern hemisphere (SH) mid and high latitudes ($55° - 85°$ N/S).





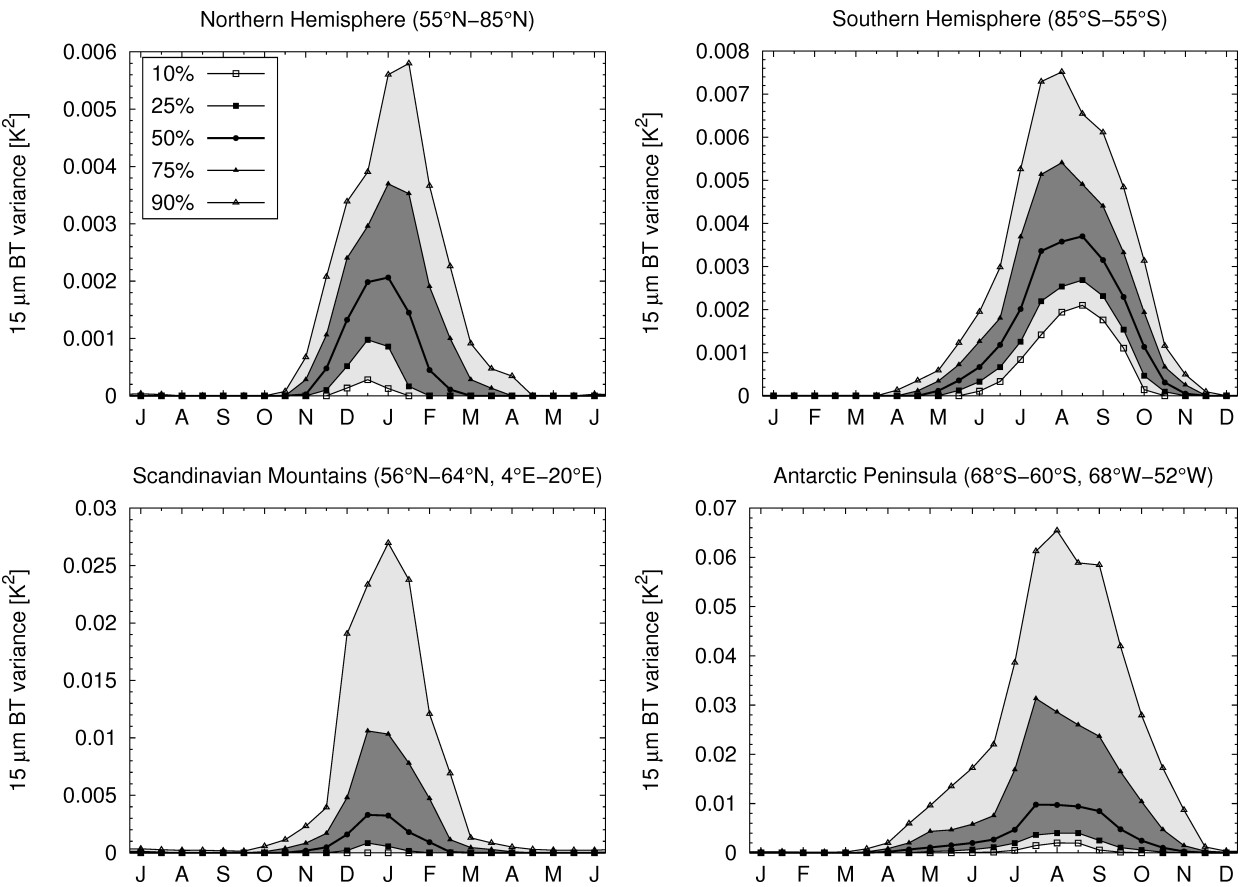

**Figure 6.** Seasonal cycle of gravity wave activity at northern (top, left) and southern (top, right) mid and high latitudes from AIRS observations in 2003 – 2012. The gravity wave activity at local hotspots such as the Scandinavian Mountains (bottom, left) and the Antarctic Peninsula (bottom, right) is also shown. Gravity wave activity is measured in terms of the 10, 25, 50, 75, and 90% quantiles of the 15 $\mu$m brightness temperature variances within 30-day time windows. Please note different scales on y-axis.





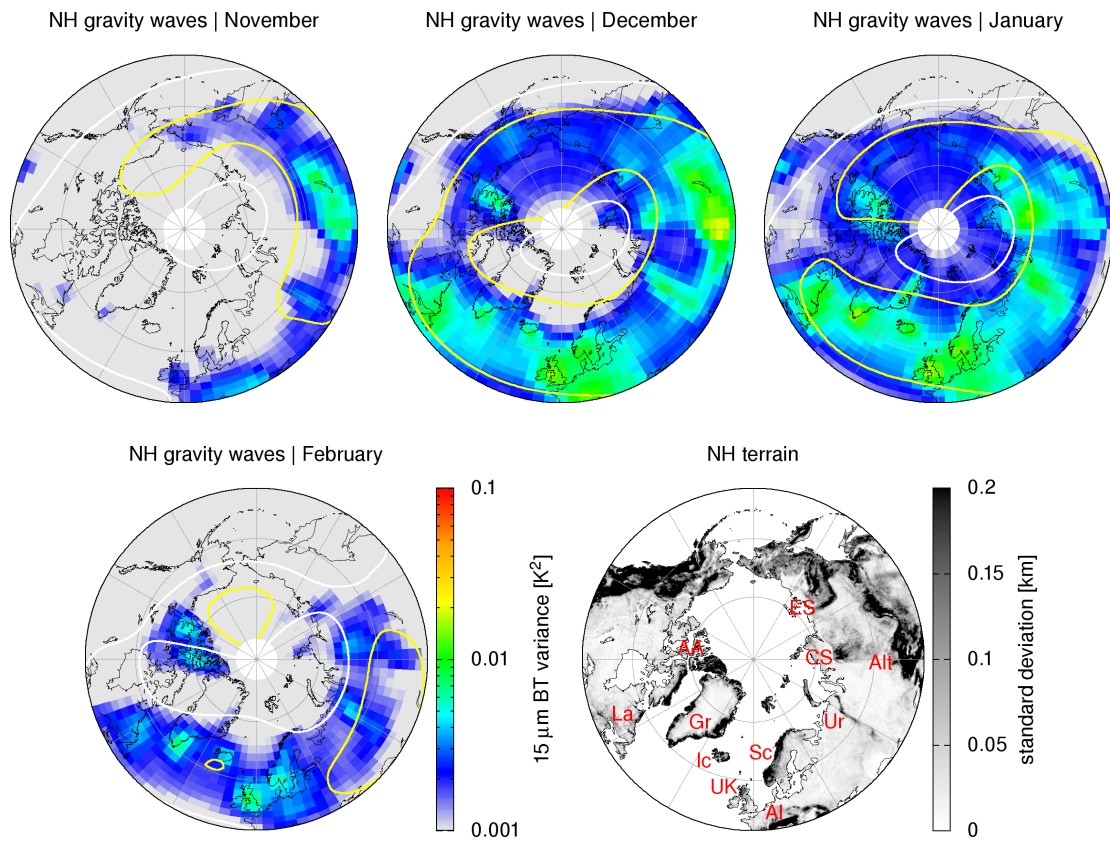

**Figure 7.** AIRS 2003 – 2012 monthly mean 15 $\mu$m brightness temperature variances from November to March in the northern hemisphere. Contour lines show 30 hPa ERA-Interim zonal winds at levels of 10 m s$^{-1}$ (white), 20 m s$^{-1}$ (yellow), 40 m s$^{-1}$ (orange), and 60 m s$^{-1}$ (red). Terrain height standard deviations on a 0.25° × 0.25° horizontal grid (bottom, right) indicate source regions of mountain waves. Red labels in the terrain map indicate the locations of the Arctic Archipelago (AA), Labrador (La), Greenland (Gr), Iceland (Ic), the United Kingdom (UK), the Scandinavian Mountains (Sc), the Alps (Al), the Ural Mountains (Ur), the Altai Mountains (Alt), the Central Siberian Plateau (CS), and the East Siberian Plateau (ES).





**Figure 8.** Same as Fig. 7, but for the southern hemisphere and the months from May to October. Red labels in the terrain map indicate the locations of Marie Byrd Land (MB), the Andes (An), the Antarctic Peninsula (AP), South Georgia (SG), Enderby Land and Mac Robertson Land (EM), the Kerguelen Islands (Ke), and the Transantarctic Mountains (TM).





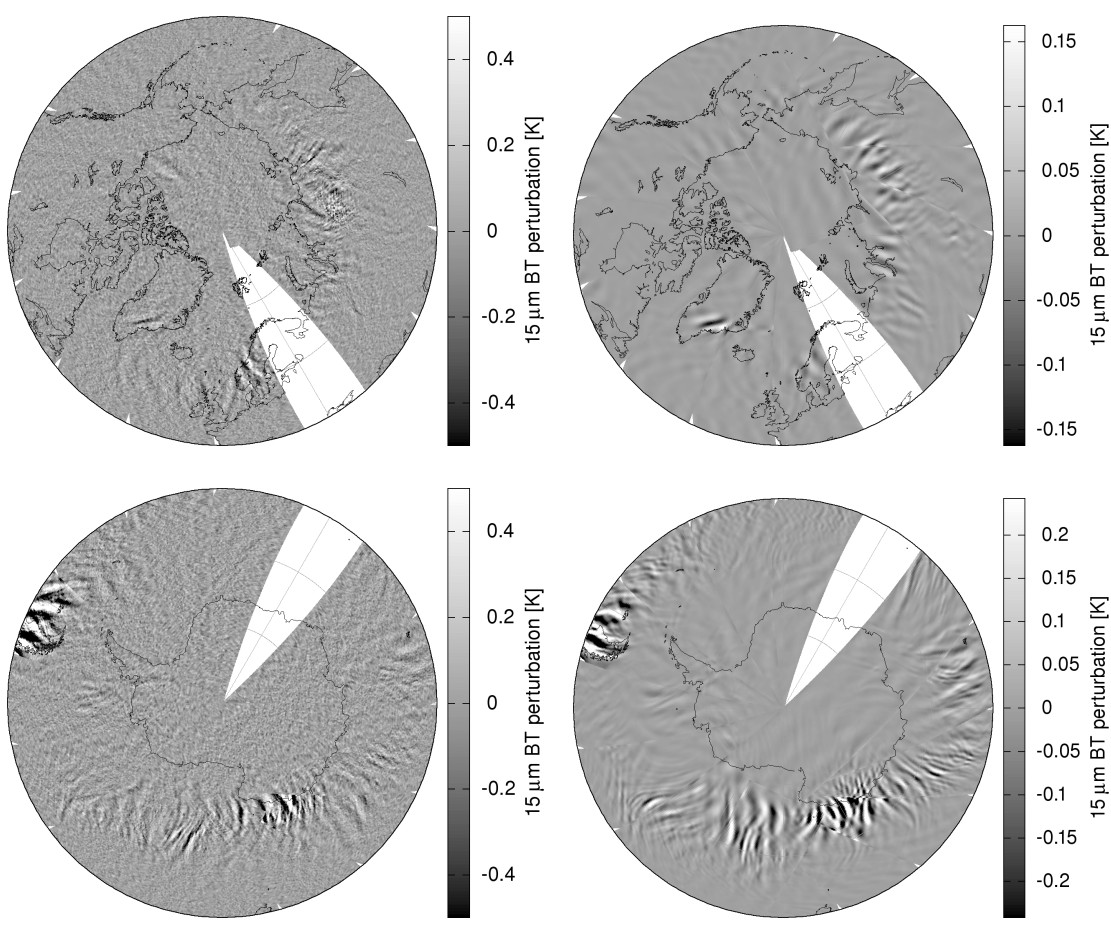

**Figure 9.** AIRS measurements of 15 μm brightness temperature perturbations (left) and corresponding simulations based on ECMWF operational analysis temperatures (right). Real measurements took place on 11 December 2003, 12:00 – 24:00 UTC in the northern hemisphere (top) and 22 July 2011, 00:00 – 12:00 UTC in the southern hemisphere (bottom). Simulated measurements are based on synoptic data at 18:00 and 06:00 UTC, respectively. Color bar ranges are scaled individually for better comparisons of the wave patterns.





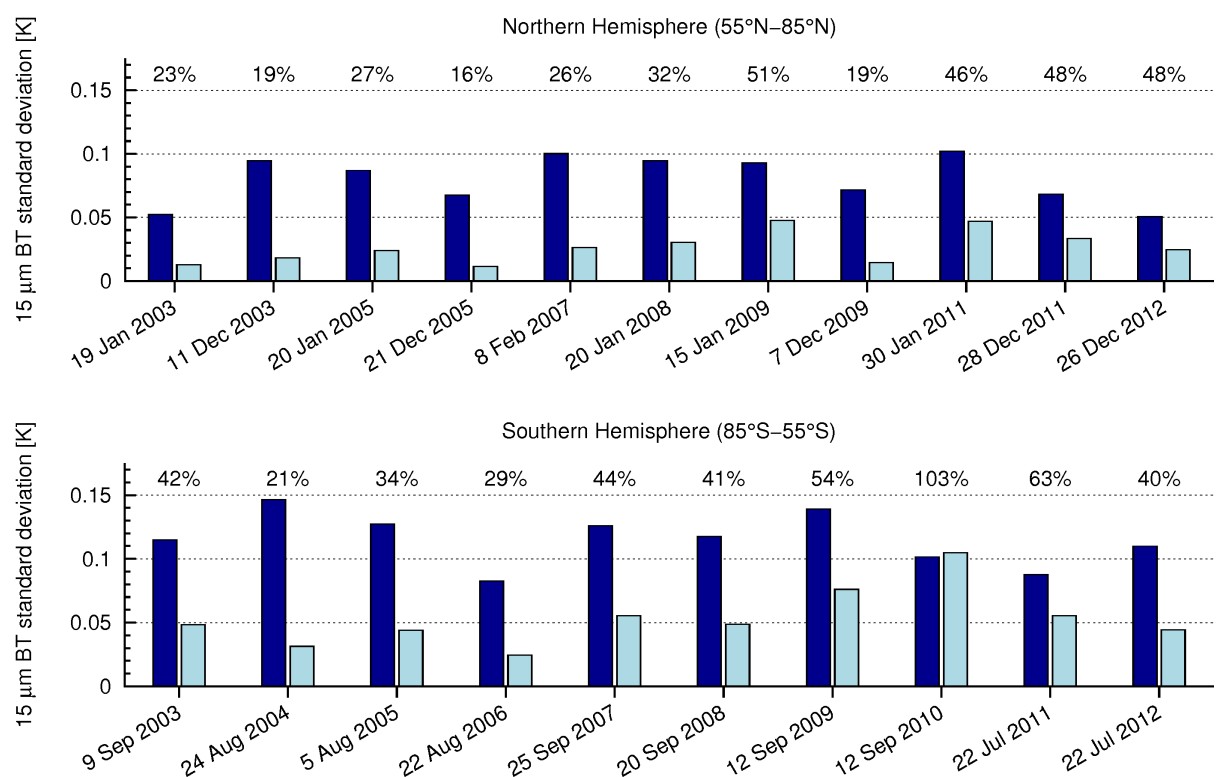

**Figure 10.** Peak events of gravity wave activity in individual 2003–2012 northern and southern hemisphere winter seasons. The 15 μm brightness temperature standard deviations were inferred from real AIRS measurements (dark blue) or simulations based on ECMWF operational analysis data (light blue). Percentage numbers at the top of the plots indicate the relative differences of the simulated data with respect to the measurements.







**Figure 11.** Examples of gravity wave-induced PSC formation events in the northern hemisphere (top) and southern hemisphere (bottom). Events are indicated by black boxes. The blue shading shows detrended and noise-corrected AIRS 15 μm brightness temperature variances. Symbols show MIPAS PSC observations (ice: blue squares, STS: red circles, NAT: green triangles, unclassified: black squares, no detections: dots) at 450–550 K potential temperature. Contour lines indicate PSC existence temperatures ($T_{ice}$: blue, $T_{STS}$: red, $T_{NAT}$: green, $T_{NAT}+3$ K: yellow) and the Montgomery stream function (gray) at the 500 K isentropic level.