# Peer review of "A decadal satellite record of gravity wave activity in the lower stratosphere to study polar stratospheric cloud formation"

_Atmospheric Chemistry and Physics, 2016_

## Referee Comment (RC1) · Anonymous Referee #1 · 12 Oct 2016

This paper is technically very strong, was written extremely well and provides new information. But, I think that the title is misleading given that the vast amount of work in this document discusses the gravity wave activity satellite record and its comparison with the ECMWF operational analysis. I do understand that when presenting a new dataset that it is vitally important to understand its strengths and weaknesses and this has been done very well in this work. But, the connection of the gravity wave activity data derived from AIRS to the formation of polar stratospheric clouds (observed by MIPAS) is extremely qualitative and appears to have been added to demonstrate future application possibilities rather than being a central part of the work. Thus, I would suggest that this paper be re-focussed so that the title and introduction better reflect

the main focus of the work or extra analysis is completed which helps to strengthen the relationship between the AIRS wave activity data and the MIPAS PSC observations. I should state that I do think this is a strong paper and is definitely worthy of publication, but at present I feel needs major revision because of this point. I identify a number of suggestions for potential improvements below.

**Suggestions: Figure 5 and 6 and corresponding discussion on Page 6 and 7:** I thought that there was a significant opportunity here to talk about the interannual and intraseasonal variability in more detail and perhaps extend the analysis. Given the discussion on temperature and wind dependencies in the rest of the work, I think adding corresponding time series of the mean 30hPa winds from ERA-interim and brightness temperatures from AIRS might be useful in aiding the interpretation of these variations. These additions might also help shows how the temperature variance varies relative to the potential timing of temperatures below PSC thresholds. The impact of features such as Sudden Stratospheric Warmings on the linkage of PSC formation to gravity waves in the Northern hemisphere might also be highlighted/explored.

**Figure 7 and 8:** Is the 2004-2012 monthly mean really representative? I would guess that variability would be rather high in the Northern hemisphere (also supported in paper) and wondered whether displaying a sample of individual years might be instructive?

**Figure 10 and corresponding text:** I understand that the operational ECMWF analyses varies, but I wondered whether some colour coding related to major changes, such as large changes in spatial resolution within the analyses might help the interpretation of Figure 10.

**Identification of PSC impacted by gravity waves:** The examples in Figure 11 seem to identify PSC using MIPAS and highlights them as anomalous when they exist in a region where the temperature is above the corresponding temperature threshold and also occur where they are co-located with sizable gravity wave temperature variance.

If this is not the method used this should be stated more clearly in the document and some statistics presented. If this is the methodology, then I would suggest some statistics would still be useful. In particular, what is the quantity of PSCs identified using MIPAS that exist in regions where temperature data would identify that they should not exist (i.e. temperatures above $T_{NAT}$ formation threshold or similar) and what is the corresponding AIRS temperature variance distribution for these cases. By comparing the distribution from that subset with the distribution data in Figure 6 you could provide a nice quantification of the importance of gravity wave activity on PSC formation.

**Minor points: Page 2 Line 16:** Earlier work by Wu and Jiang (2002), Shibata et al. (2003) and Baumgaertner and McDonald (2007) might be worthwhile additions to your list.

**Page 10 Sentence starting on Line 21:** I think Alexander et al. (2011) and Alexander et al. (2013) might be worthwhile citing at this point since they highlighted the importance of advection of PSCs away from the region where the temperature perturbations were observed.

**Figure 11:** I think this figure (as the major proof of the connection between anomalous PSC formation and wave activity) could be improved. At present, it is complex enough that I have to squint to see what is happening and is overall rather busy.First, I think the coloured markers are not used effectively. It would be helpful to colour code the markers based on whether the corresponding temperatures are above or below the corresponding temperature thresholds and use only the shape to identify different PSC types (or perhaps vice versa). I would also remove the 'no detection' dots as they do not really add value and clutter the diagram. I also think that the streamlines, while useful, crowd the diagram – so perhaps using less of them. I would also make the black boxes highlighting the areas of interest more defined.

**References:**

Alexander, S. P., et al. (2011). "The effect of orographic gravity waves on Antarctic polar

stratospheric cloud occurrence and composition." Journal of Geophysical Research-Atmospheres 116.

Alexander, S. P., et al. (2013). "Quantifying the role of orographic gravity waves on polar stratospheric cloud occurrence in the Antarctic and the Arctic." Journal of Geophysical Research-Atmospheres 118(20): 15.

Baumgaertner, A. J. G. and A. J. McDonald (2007). "A gravity wave climatology for Antarctica compiled from Challenging Minisatellite Payload/Global Positioning System (CHAMP/GPS) radio occultations - art. no. D05103." Journal of Geophysical Research-Atmospheres 112(D5): 5103-5103.

Shibata, T., et al. (2003). "Antarctic polar stratospheric clouds under temperature perturbation by nonorographic inertia gravity waves observed by micropulse lidar at Syowa Station." Journal of Geophysical Research-Atmospheres 108(D3).

Wu, D. L. and J. H. Jiang (2002). "MLS observations of atmospheric gravity waves over Antarctica." Journal of Geophysical Research-Atmospheres 107(D24).

---

## Referee Comment (RC2) · Anonymous Referee #2 · 25 Oct 2016

The paper describes an accurate analysis of the gravity wave activity in the lower stratosphere retrieved from AIRS for the northern and southern hemispheres. A new methodology (detrended and noise-corrected 15 micrometer brightness temperature variances, which are calculated from AIRS channels that are most sensitive to temperature fluctuations at about 17 – 32 km altitude) was developed to introduce the new data set which is presented in this paper. The results of this part of the paper are convincing and the authors put a lot of effort to weight their results in terms of error analysis, observational filter etc. Thus, the formulated goal to provide "the new AIRS data set to identify local hotspots and sources of gravity wave activity, to characterize its seasonal cycle at northern and southern mid and high latitudes, and to analyze

correlations with stratospheric background winds" is well done.

The PSC part is much weaker. Especially, the discussion related to Fig. 11 is only partialy convincing.

First of all, in contrast to Section 4, here ERA Interim data are used to specify the atmospheric background. This certanily leads to smooth temperature fields not containing any mesoscale gravity wave activity. I suppose, this was done by purpose. But why? As shown in this paper ECMWF operational analyses and forecasts are reliable to detect gravity wave activity.

Second, the assigment of PSC observations from MIPAS to the AIRS gravity wave activity is somehow strange for me. For example, take Fig. 11 for 25 Jan 2007: Ice PSCs are observed over Scandinavia, enhanced gravity wave activity over Greenland. The respective text reads: "On 25 January 25 2007, MIPAS detected ice PSCs over Scandinavia as well as Germany and Poland at synoptic-scale temperatures up to $6 - 9K$ above Tice. These detections are located downstream of Greenland, where strong gravity wave activity was present at the east and north coast according to the AIRS observations. Weak gravity wave activity is also visible over the Scandinavian Mountains."

The ice PSCs over Germany and Poland are not marked in the plot. But this is minor. Do the authors suggest that the mountain waves over Greenland formed the ice PSCs some 1000 km downstream over Scandinavia? I think so as a quite similar text passage is given for the 7 Jan 2011 case. I would suggest to either run simple backward trajectories to see if Tice was reached on some stage or to use different diagnostics to detect mesoscale gravity wave activity from the ECMWF operational data above the actual observation location. Recently, Khaykin, S. M., A. Hauchecorne, N. Mzé, and P. Keckhut, 2015: Seasonal variation of gravity wave activity at midlatitudes from 7 years of COSMIC GPS and Rayleigh lidar temperature observations, Geophys. Res. Lett., 42, 1251–1258, doi:10.1002/2014GL062891 used a threshold value of the horizontal

divergence to relate their observed gravity wave-induced temperature perturbations to the atmospheric state simulated by the ECMWF. Another possibility would be to use CALIPSO data if available for the time periods to verify the extent, depth, and heights of the detected ice PSCs.

As I wrote before, Section 5 needs careful revision.

---

## Author Comment (AC1) · 7 Dec 2016

The comment was uploaded in the form of a supplement:
http://www.atmos-chem-phys-discuss.net/acp-2016-757/acp-2016-757-AC1-supplement.pdf

---

## Author Response (AR1)

**Reply to review comments**

We thank the reviewers for the thoughtful comments and the time and efforts spent on the manuscript. Please find our point-by-point replies to the review comments below (colored in blue). A revised manuscript with tracked changes was uploaded.

**Anonymous Reviewer #1**

This paper is technically very strong, was written extremely well and provides new information. But, I think that the title is misleading given that the vast amount of work in this document discusses the gravity wave activity satellite record and its comparison with the ECMWF operational analysis. I do understand that when presenting a new dataset that it is vitally important to understand its strengths and weaknesses and this has been done very well in this work. But, the connection of the gravity wave activity data derived from AIRS to the formation of polar stratospheric clouds (observed by MIPAS) is extremely qualitative and appears to have been added to demonstrate future application possibilities rather than being a central part of the work. Thus, I would suggest that this paper be re-focused so that the title and introduction better reflect the main focus of the work or extra analysis is completed which helps to strengthen the relationship between the AIRS wave activity data and the MIPAS PSC observations. I should state that I do think this is a strong paper and is definitely worthy of publication, but at present I feel needs major revision because of this point. I identify a number of suggestions for potential improvements below.

In this revision we tried to improve the PSC analysis following the comments provided by both reviewers. We added new material regarding gravity wave-induced PSC formation to Sections 3 and 5 of the paper. Although the AIRS gravity wave data set itself is the main topic of the paper, we think that is worthwhile pointing out that this data set was specifically optimized to study PSC formation (in particular in terms of vertical coverage of the selected AIRS channels) and suggest to keep the title as is for the time being.

Suggestions: Figure 5 and 6 and corresponding discussion on Page 6 and 7: I thought that there was a significant opportunity here to talk about the interannual and intraseasonal variability in more detail and perhaps extend the analysis. Given the discussion on temperature and wind dependencies in the rest of the work, I think adding corresponding time series of the mean 30hPa winds from ERA-interim and brightness temperatures from AIRS might be useful in aiding the interpretation of these variations. These additions might also help shows how the temperature variance varies relative to the potential timing of temperatures below PSC thresholds. The impact of features such as Sudden Stratospheric Warmings on the linkage of PSC formation to gravity waves in the Northern hemisphere might also be highlighted/explored. We followed these suggestions and added time series of ERA-Interim 30 hPa zonal winds and 60 hPa temperatures to Fig. 5. We decided to show ERA-Interim temperatures instead of AIRS brightness temperatures because we consider them to be more representative for the PSC analysis. The AIRS brightness temperatures cover a large altitude range and typically show a positive offset (5-10 K) compared with 60 hPa temperatures. They can therefore not be compared directly to PSC formation temperatures. Furthermore, to achieve consistency with Fig. 6, we decided to add time series for the Antarctic Peninsula and the Scandinavian Mountains. In Fig. 6 we added monthly mean occurrence frequencies of 60 hPa temperatures falling below  $T_{NAT}$ , which allows us to compare the timing of gravity wave activity with the timing of synoptic scale PSC occurrence. We also added monthly mean occurrence frequencies of 30 hPa zonal winds exceeding a threshold of 20 m/s. As discussed in the paper, this is a requirement to be able to detect gravity wave activity with AIRS due to the observational filter effect. The discussion of Figs. 5 and 6 in Section 3 of the paper was revised accordingly. We also added a note that SSWs can be identified in the time series, linked with a significant decrease in observed gravity wave activity.

Figure 7 and 8: Is the 2004-2012 monthly mean really representative? I would guess that variability would be rather high in the Northern hemisphere (also supported in paper) and wondered whether displaying a sample of individual years might be instructive?

To clarify we added the following statement in Section 3: "As there is large variability of gravity wave activity, in particular in the northern hemisphere, we calculated the standard errors of the monthly means shown in Figs. 7 and 8 to assess if they are representative. We found standard errors of 29-39% for the months from November to February for the northern hemisphere and 18-21% for the months from May to October for the southern hemisphere."

Figure 10 and corresponding text: I understand that the operational ECMWF analyses varies, but I wondered whether some colour coding related to major changes, such as large changes in spatial resolution within the analyses might help the interpretation of Figure 10.

We added color coding to Fig. 10 as suggested. This helps with the interpretation as it shows that improvements in the resolution of the ECMWF operational analysis tend to go along with improvements in the representation of explicitly resolved gravity waves.

Identification of PSC impacted by gravity waves: The examples in Figure 11 seem to identify PSC using MIPAS and highlights them as anomalous when they exist in a region where the temperature is above the corresponding temperature threshold and also occur where they are co-located with sizable gravity wave temperature variance. If this is not the method used this should be stated more clearly in the document and some statistics presented. If this is the methodology, then I would suggest some statistics would still be useful. In particular, what is the quantity of PSCs identified using MIPAS that exist in regions where temperature data would identify that they should not exist (i.e. temperatures above  $T_{NAT}$  formation threshold or similar) and what is the corresponding AIRS temperature variance distribution for these cases. By comparing the distribution from that subset with the distribution data in Figure 6 you could provide a nice quantification of the importance of gravity wave activity on PSC formation.

The method used to identify anomalous (i. e., potentially GW-induced) PSC detections by MIPAS was recapitulated correctly by the reviewer. We agree that counting statistics of these events are very interesting. We intend to analyze these statistics in more detail as part of a new study that is currently in preparation.

Minor points: Page 2 Line 16: Earlier work by Wu and Jiang (2002), Shibata et al. (2003) and Baumgaertner and McDonald (2007) might be worthwhile additions to your list.

We added these references.

Page 10 Sentence starting on Line 21: I think Alexander et al. (2011) and Alexander et al. (2013) might be worthwhile citing at this point since they highlighted the importance of advection of PSCs away from the region where the temperature perturbations were observed.

We added a corresponding statement and the references.

Figure 11: I think this figure (as the major proof of the connection between anomalous PSC formation and wave activity) could be improved. At present, it is complex enough that I have to squint to see what is happening and is overall rather busy.First, I think the coloured markers are not used effectively. It would be helpful to colour code the markers based on whether the corresponding temperatures are above or below the corresponding temperature thresholds and use only the shape to identify different PSC types (or perhaps vice versa). I would also remove the no detection dots as they do not really add value and clutter the diagram. I also think that the streamlines, while useful, crowd the diagram so perhaps using less of them. I would also make the black boxes highlighting the areas of interest more defined.

We improved Fig. 11 following these suggestions.

**Anonymous Reviewer #2**

The paper describes an accurate analysis of the gravity wave activity in the lower stratosphere retrieved from AIRS for the northern and southern hemispheres. A new methodology (detrended and noise-corrected 15 micrometer brightness temperature variances, which are calculated from AIRS channels that are most sensitive to temperature fluctuations at about 17–32 km altitude) was developed to introduce the new data set which is presented in this paper. The results of this part of the paper are convincing and the authors put a lot of effort to weight their results in terms of error analysis, observational filter etc. Thus, the formulated goal to provide "the new AIRS data set to identify local hotspots and sources of gravity wave activity, to characterize its seasonal cycle at northern and southern mid and high latitudes, and to analyze correlations with stratospheric background winds" is well done.

The PSC part is much weaker. Especially, the discussion related to Fig. 11 is only partially convincing.

In this revision of the paper we tried to improve the PSC part, following the comments provided by both reviewers.

First of all, in contrast to Section 4, here ERA Interim data are used to specify the atmospheric background. This certainly leads to smooth temperature fields not containing any mesoscale gravity wave activity. I suppose, this was done by purpose. But why? As shown in this paper ECMWF operational analyses and forecasts are reliable to detect gravity wave activity.

In Section 5 of the paper we focus on a combined analysis of AIRS observations of gravity waves and MIPAS observations of PSCs. The ERA-Interim data are intended to be used only to provide supplementary information in terms of PSC formation temperatures and Montgomery streamfunctions, but should not be used to infer information on gravity waves directly. The same approach was used by *Spang et al.* (2016). However, we had left this frame and started to discuss wave signatures in the ERA-Interim data on page 11, lines 5-10 of the manuscript. We revised this part. Note that we found that although the ECMWF operational analysis is often showing gravity wave activity in the correct locations, the wave amplitudes are significantly underestimated. Furthermore, the skills of the analysis change over time. The ERA-Interim data provide a more consistent picture.

Second, the assignment of PSC observations from MIPAS to the AIRS gravity wave activity is somehow strange for me. For example, take Fig. 11 for 25 Jan 2007: Ice PSCs are observed over Scandinavia, enhanced gravity wave activity over Greenland. The respective text reads: "On 25 January 25 2007, MIPAS detected ice PSCs over Scandinavia as well as Germany and Poland at synoptic-scale temperatures up to 6-9K above  $T_{ice}$ . These detections are located downstream of Greenland, where strong gravity wave activity was present at the east and north coast according to the AIRS observations. Weak gravity wave activity is also visible over the Scandinavian Mountains."

The ice PSCs over Germany and Poland are not marked in the plot. But this is minor. Do the authors suggest that the mountain waves over Greenland formed the ice PSCs some 1000 km downstream over Scandinavia? I think so as a quite similar text passage is given for the 7 Jan 2011 case. I would suggest to either run simple backward trajectories to see if  $T_{ice}$  was reached on some stage or to use different diagnostics to detect mesoscale gravity wave activity from the ECMWF operational data above the actual observation location. Recently, Khaykin, S. M., A. Hauchecorne, N. Mz, and P. Keckhut, 2015: Seasonal variation of gravity wave activity at midlatitudes from 7 years of COSMIC GPS and Rayleigh lidar temperature observations, Geophys. Res. Lett., 42, 12511258, doi:10.1002/2014GL062891 used a threshold value of the horizontal divergence to relate their observed gravity wave-induced temperature perturbations to the atmospheric state simulated by the ECMWF. Another possibility would be to use CALIPSO data if available for the time periods to verify the extent, depth, and heights of the detected ice PSCs.

We agree that cross checks of the case studies presented in Fig. 11 are helpful. Therefore we added two new figures showing CALIPSO data and trajectory calculations for two of the case studies (25 January 2007 and 8 July 2008), as representative examples. Two new paragraphs in Section 5 provide a discussion of the results. From the trajectory calculations we concur that ice PSC formation over Greenland on 25 January 2007 is unlikely (making gravity waves over Scandinavia a more likely source) and revised the text accordingly. We removed the statement regarding PSC detections over Germany and Poland. We also revised the interpretation of the 7 January 2011 case.

As I wrote before, Section 5 needs careful revision.

Section 5 was revised according to the comments of both reviewers.

**Additional changes**

We replaced the references (Spang et al., 2004; Höpfner et al., 2006a,b; Spang et al., 2012, 2014) by (Spang et al., 2004, 2005; Höpfner et al., 2006a,b; Spang et al., 2012) on page 10, line 3.

[revised manuscript text omitted]